# Two waves of pro-inflammatory factors are released during the influenza A virus (IAV)-driven pulmonary immunopathogenesis

**Junsong Zhang**[1,2]☯, **Jun Liu**[1]☯, **Yaochang Yuan**[1], **Feng Huang**[3], **Rong Ma**[1], **Baohong Luo**[1], **Zhihui Xi**[4], **Ting Pan**[1,5], **Bingfeng Liu**[1], **Yiwen Zhang**[1], **Xu Zhang**[1], **Yuewen Luo**[1], **Jin Wang**[6], **Meng Zhao**[6], **Gen Lu**[3], **Kai Deng**[1]*, **Hui Zhang**[1]*

**1** Institute of Human Virology, Key Laboratory of Tropical Disease Control of Ministry of Education, Guangdong Engineering Research Center for Antimicrobial Agent and Immunotechnology, Zhongshan School of Medicine, Sun Yat-sen University, Guangzhou, Guangdong, China, **2** Guangdong Provincial People's Hospital, Guangdong Academy of Medical Sciences, Guangzhou, Guangdong, China, **3** Department of Respiratory Diseases, Guangzhou Women and Children Hospital, Guangzhou, Guangdong, China, **4** Qianyang Institute of Biomedical Research, Guangzhou, Guangdong, China, **5** School of Medicine, Sun Yat-sen University, Guangzhou, Guangdong, China, **6** Department of Pathophysiology, Zhongshan School of Medicine, Sun Yat-sen University, Guangzhou, Guangdong, China

☯ These authors contributed equally to this work.
* dengkai6@mail.sysu.edu.cn (KD); zhangh92@mail.sysu.edu.cn (HZ)

**Data Availability Statement:** All relevant data are within the manuscript and its Supporting Information files.

## Abstract

Influenza A virus (IAV) infection is a complicated process. After IAVs spread to the lung, extensive pro-inflammatory cytokines and chemokines are released, which largely determine the outcome of infection. Using a single-cell RNA sequencing (scRNA-seq) assay, we systematically and sequentially analyzed the transcriptome of more than 16,000 immune cells in the pulmonary tissue of infected mice, and demonstrated that two waves of pro-inflammatory factors were released. A group of IAV-infected PD-L1+ neutrophils were the major contributor to the first wave at an earlier stage (day 1–3 post infection). Notably, at a later stage (day 7 post infection) when IAV was hardly detected in the immune cells, a group of platelet factor 4-positive (Pf4+)-macrophages generated another wave of pro-inflammatory factors, which were probably the precursors of alveolar macrophages (AMs). Furthermore, single-cell signaling map identified inter-lineage crosstalk between different clusters and helped better understand the signature of PD-L1+ neutrophils and Pf4+-macrophages. Our data characteristically clarified the infiltrated immune cells and their production of pro-inflammatory factors during the immunopathogenesis development, and deciphered the important mechanisms underlying IAV-driven inflammatory reactions in the lung.

## Author summary

Influenza A virus (IAV) infections cause acute respiratory disease in many species, including human, mammals and birds, and are responsible for a number of pandemics among humans, resulting in substantial morbidity and mortality. High morbidity and mortality of IAV-driven pneumonia reflects the deficient immunity of the hosts against IAV

**Funding:** The present study was supported by the National Special Research Program of China for Important Infectious Diseases (2018ZX10302103, 2017ZX10202102-003, and 2018ZX10101004003001), Important Key Program of Natural Science Foundation of China (81730060), the National Natural Science Foundation of China (81701990), and the Joint-innovation Program in Healthcare for Special Scientific Research Projects of Guangzhou (201803040002). The funders had no role in study design, data collection and analysis, decision to publish, or preparation of the manuscript.

**Competing interests:** The authors have declared that no competing interests exist.

infection, and the inefficiency of available prevention and treatment strategies. Thus, in depth exploration of IAV pathogenesis is necessary. In our study, using the transverse (cells to cells) and longitudinal (day to day) analysis of immune cells in the lung, we monitored the whole immunopathogenesis during IAV infection, and identified several cell types as contributors for the release of pro-inflammatory factors. Therefore, our study potentially provides new therapeutic targets for IAV treatment.

## Introduction

Aberrant pulmonary immune responses correlate with the pathogenesis of multiple human respiratory viral infections, including IAV infection [1]. Immune responses in the lung tissue include both antiviral and inflammatory factors, which play crucial roles in host protection and immunopathogenesis [2, 3]. Through the integrated action of different pro-inflammatory factors, different immune cells are recruited into the airway. CC chemokines (like Ccl2, Ccl7, and Ccl8) and CXC chemokines (like Cxcl2 and Cxcl12) are important for the recruitment of leukocytes into the microenvironment of the airway. Infected monocytes and macrophages are the main contributors to the rapid production of pro-inflammatory and chemotactic cytokines, which lead to the enhanced migration of leukocytes and result in an effective defense against viral invasion. However, elevated cytokine and chemokine production has been associated with a poor clinical outcome [4]. Studies have highlighted a correlation between IL-6, IL-1, and TNF-α levels and the severity of disease symptoms [5, 6, 2]. Though these infiltrated cells are required for host protection and recovery, they can also exacerbate the immune injury to the lung and worsen clinical symptoms. The activation of neutrophils was recently reported associated with the most severe and acute infection of IAV in patients [7], and old mice infected with IAV would induced excessive levels of neutrophils and higher levels of cytokines [8], indicating that neutrophils have important roles in the IAV-driven immunopathogenesis Because of the double-edged sword role played by these infiltrated immune cells and the cytokines/chemokines they produced, it is necessary to further explore immune reaction profiles of the lung at different time points during IAV infection.

Alveolar macrophages (AMs) are critical for lung homeostasis and immune responses to pathogens [9]. As tissue-resident macrophages, AMs can self-maintain in local without the contribution of bone marrow (BM)-derived monocyte under normal condition [10]. TGF-bR signaling can up-regulate the expression of PPAR-γ, a signature transcription factor that is essential for the development of AMs [11]. Dramatically reduction of AMs was found in the lung infected with IAVs. Of note, AMs undergo M1 and M2 polarization through different stimulation by different cytokines. During IAV infection, a large number of AMs undergo apoptosis. When viruses are eliminated by neutralization antibody or T cells, the local repopulation of AMs would be dependent on BM-derived monocytes in a short time [12]. However, the long-time recovery of AMs depended on the local proliferation of the tissue resident AMs. The mechanism how the AMs return to steady state during IAV-derived lung damage remained to be clarified.

Single-cell RNA sequencing (scRNA-seq) has been applied to investigate the immune system under physiological and pathological conditions [13–16]. It allows detailed understanding of the complicate immune system at single-cell resolution [17–19]. In particular, scRNA-seq is a powerful tool for defining viral target cells, as it is convenient for analyzing the viral mRNAs and host signature genes in a single cell [20–23]. In addition, it can precisely examine the patterns of cytokine release in each immune cell and inter-lineage crosstalk in single-cell with the

ligand and receptor interaction map [24, 25]. In this study, we analyzed >16,000 immune cells in the lung tissue isolated from mice infected with IAV at different time points post infection, and scRNA-seq enabled us to clarify the complicated immune responses in the lung tissue across the whole course of IAV-driven pneumonia.

## Results

### Identification of cell clusters in the lung during IAV infection

To investigate the whole immune cell populations in the lung during IAV infection, we collected total suspended cells of the lung tissue from C57BL/6 mice uninfected (day 0, 2762 cells) or infected with A/PR/8/34 (H1N1) virus at 5 time points including day 1 (2185 cells), day 3 (3074 cells), day 5 (2526 cells), day 7 (2572 cells), and day 12 (3305 cells) (3 mouse each group), for the droplet-based scRNA-seq transcriptional profiles using the 10x Chromium platform (Fig 1A and S1 Table). Any cell with less than 200 genes or more than 30% of mitochondrial unique molecular identifier (UMI) counts was filtered out, and only genes with at least one UMI count detected in at least one cell were used for further analysis. The scRNA-seq profiles after the quality control were aggregated and analyzed using CellRanger software, which can provide stable and accurate clustering solutions for 10x Genomics scRNA-seq data [26]. The sequencing quality control showed that the six samples from six time points during virus infection were qualified for further scRNA-seq analysis (S1A Fig and S1 Table). Graph-based clustering was run using t-distributed Stochastic Neighbor Embedding (tSNE) to group cells together that have similar expression profiles, and to build a sparse nearest-neighbor graph without pre-specification of the number of clusters.

Based on the tSNE dimensionality reduction and unsupervised cell clustering, we identified 18 distinct cell clusters named as C1-C18 based upon their total cell numbers, which expressed unique transcriptional profiles and sequentially occurred at different time points (Fig 1B, S1B Fig and S2 Table). Pairwise Pearson correlations between each cluster were calculated based on the mean expression of each gene across all cells in the cluster for hierarchical clustering, showing the distinct relationship among different clusters (S1C Fig and S3 Table). We also found that the changes of several major clusters during IAV infection with FACS analysis were similar with that in scRNA-seq data (S1D Fig and S2 Table), which further confirmed the scRNA-seq data. To identify genes that were enriched in a specific cluster, the mean expression of each gene was calculated across all cells in the cluster and the log2 fold-change of differentially expressed genes was calculated relative to the other clusters. Some significant genes (Log2 fold change >1, *P*-value <0.01, Benjamini-Hochberg adjusted) and genes with high expression of the known markers of major cell types were shown in Fig 1C. For example, many cells in cluster 5 (denoted C5) showed high expression of SiglecF and CD11c (Itgax), and were labeled as pulmonary alveolar macrophages (AMs) (S2A Fig). The above results, combined with further principal component analysis demonstrated that the immune cells in the lung comprised all major immune lineages, and the clusters were mainly comprised by monocyte/macrophage/DC-lineage (C1, C5, C8, C7, C10, and C17), lymphocyte-lineage (C2, C3, C4, C6, C11, and C12), granulocyte-lineage (C13, C14 and C16), erythrocyte-lineage (C9 and C15), and epithelial cell-lineage (C18) (S2 Fig). The heterogenous components of cells in the lung during IAV infection highlight the necessity of single-cell analysis for dissecting the IAV-related immune cells in the lung in detail.

IAV infection initiates in the respiratory tract and spreads in the lung, which triggers widespread pulmonary immune responses. We fitted the cells from 18 clusters into different samples that were collected at different times. The distribution patterns of the 18 clusters sequentially changed at different time points after IAV infection (S3 Fig). Massive changes to

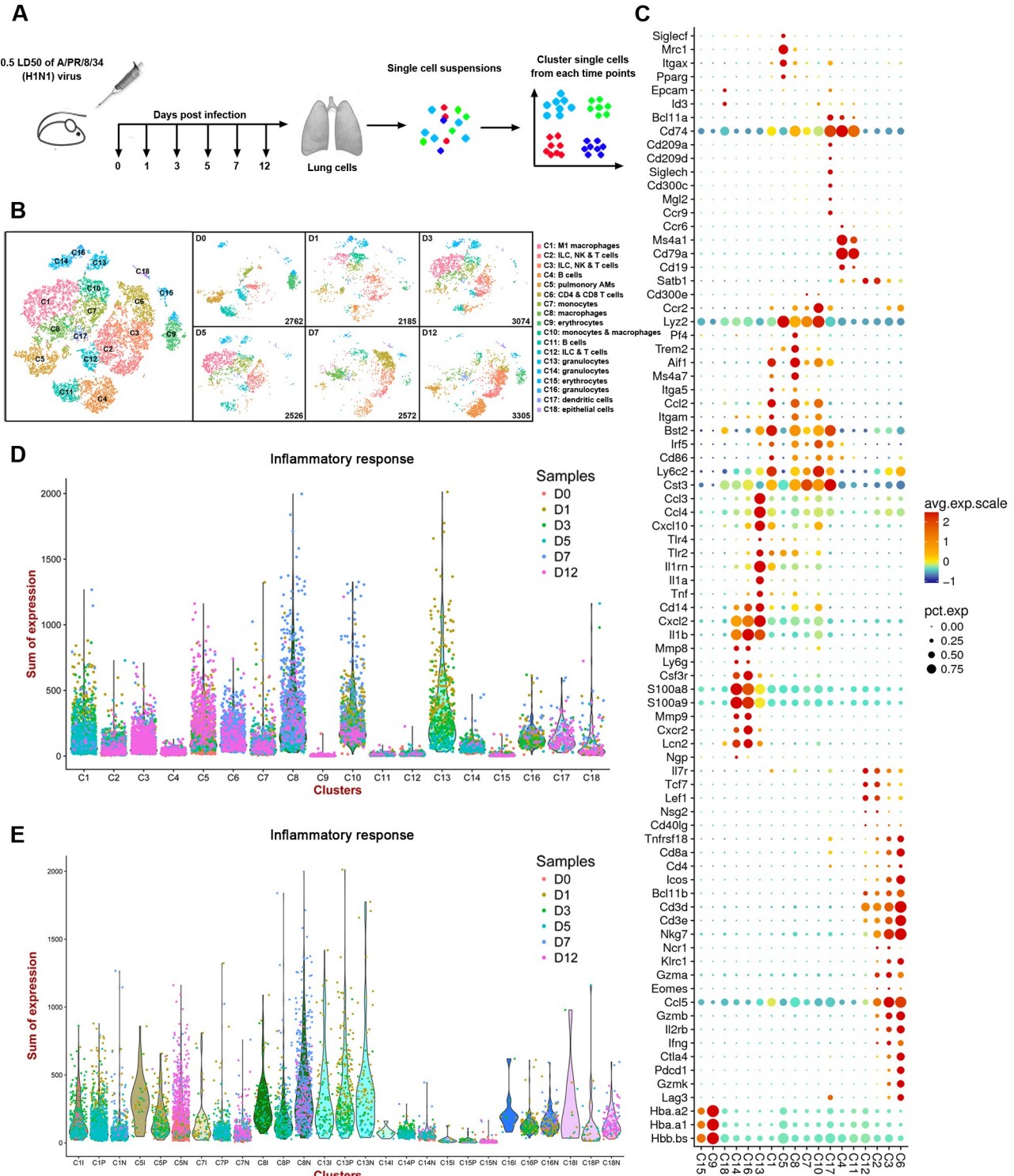

**Fig 1. Single-cell profiling of cell populations in the lung during IAV infection.** (**A**) Overview of the study design. Four mice for each group were infected with influenza A/PR/8/34 (H1N1) viruses and sacrificed at the indicated time points p.i.. The lungs of the four mice were mixed together as one sample for single-cell sequencing analysis. (**B**) tSNE maps displaying 16,424 suspended cells from the lung and coloured by the main cell populations based on the unsupervised graph-based, showing the formation of 18 main clusters with the cell numbers (Left panel). The scRNA-seq data of 16,424 cells from six libraries at different days p.i. were normalized by equalizing the read depth before merging using the cellranger aggregate procedure. The tSNE maps of the six maps in the right panel showing the formation of 18 main clusters form six libraries at different days p.i. The data between libraries was normalized by equalizing the read depth between libraries before merging and comparison. (**C**) Representative differentially

expressed genes (y axis) by cluster (x axis). Dot size represents the fraction of cells in the cluster that express the gene; colour indicates the mean expression (Z-score) in expressing cells, relative to other clusters. (**D**) The sum UMI counts expression of host 372 genes related to inflammatory response in different cell clusters (X-axis) (GO: 0006954).The dots indicate the cells from different clusters, coloured according to the samples. (**E**) The sum UMI counts expression of host 372 genes related to inflammatory response in different cell clusters susceptible to IAV infection (X-axis) (GO: 0006954). The dots indicate the cells from different clusters, coloured according to the samples. The cells in the clusters susceptible to IAV infection were divided into highly infected cells (I), potential or lowly infected cells (P), and undetected cells (N).

the transcriptional landscape were found between the normal lung tissue (day 0) and tissue from mice challenged with A/PR/8/34 (H1N1) viruses (day 1, 3, 5, 7, and 12). We also analyzed the expression of genes involved in the inflammatory response at different time points and in different clusters. There are 372 genes related to the GO term of inflammatory response (GO: 0006954) (S4 Fig). The inflammatory response was analyzed within various cells. C1/C5/C7/C10-monocytes and C13/C14/C16-granulocytes generated many pro-inflammatory factors at various time points (Fig 1D and S5 Fig).

Host cells infected with IAV were quantitatively identified by tracking the intracellular IAV segmented mRNAs at single-cell resolution. To better understand the infected cells in lung, the UMI counts of IAV genes were sought in single cell transcriptional data (S5 and S6 Tables), and above 5000 cells were detected containing at least one counts of IAV transcript (S6 Fig). To reduce the false positive rate of infected cells, only the cells with the highly expressed IAV genes (at least one transcript per gene per cell) were defined as highly infected cells. There were 668 cells selected with the expression of more than 8 copies of viral mRNAs. Among them, the cell number at one day p.i. is 135 cells; three days p.i. is 352 cells; five days p.i. is 180 cells; seven days p.i. is 1 cells; 12 days p.i. is 0 cells). The viral mRNA-positive cells mainly appeared in samples from day 1, day 3 and day 5 p.i., while a few RNA copies of virus genes were found at day 7 and day 12 p.i., showing that the clearance of the viruses occurred at day 7 p.i. (S7 Fig and S6 Table). Significant amount of viral mRNAs were mainly detected in the cells from 8 clusters (C1, C5, C7, C8, C13, C14, C15, C16 and C18) p.i. (S8 Fig). According to the expression counts of IAV transcripts, the cells in the clusters susceptible to IAV infection can be divided into highly infected cells (total UMI counts of viral transcripts $\geq 8$), potential or lowly infected cells (total UMI counts of viral transcripts $\geq 1$), and undetected cells (UMI counts of viral transcripts = 0) (S8 Fig). However, the cells of three types exhibited similar response to IAV in most clusters, indicating that there was no visible correlation between viral load and host response level of single cell in the clusters infected to IAV (Fig 1E). The cells under both extracellular exposure and intracellular infection can exhibit a significant response in addition to a significant bystander response [27, 28]. The uninfected or lowly infected immune cells could be activated by cytokines or chemokines from other immune cells and generate antiviral factors or other pro-inflammatory factors. These results depicted the dynamic landscape of the cells from the lung immune response during IAV infection, and described the composition of immune cells during IAV-driven pneumonia.

## PD-L1$^+$ neutrophils infected by IAV are the major contributor to the first wave of pro-inflammatory factors

Sequential transcriptional profile analysis specifically for pro-inflammatory factors revealed that there were two waves of pro-inflammatory factor productions during the whole IAV infection process. The first wave occurred during the early stage of IAV infection, and the second wave occurred after day 7 p.i. when the viral replication was no longer detectable (Fig 2A). Further study on the pro-inflammatory factors at day 1 p.i. indicated that although C1-macrophages generated some factors, such as Ccl22 and Cxcl19 and C5-pulmonary AMs generated IL18, the C13-cells are the major contributor to generating various pro-inflammatory factors,

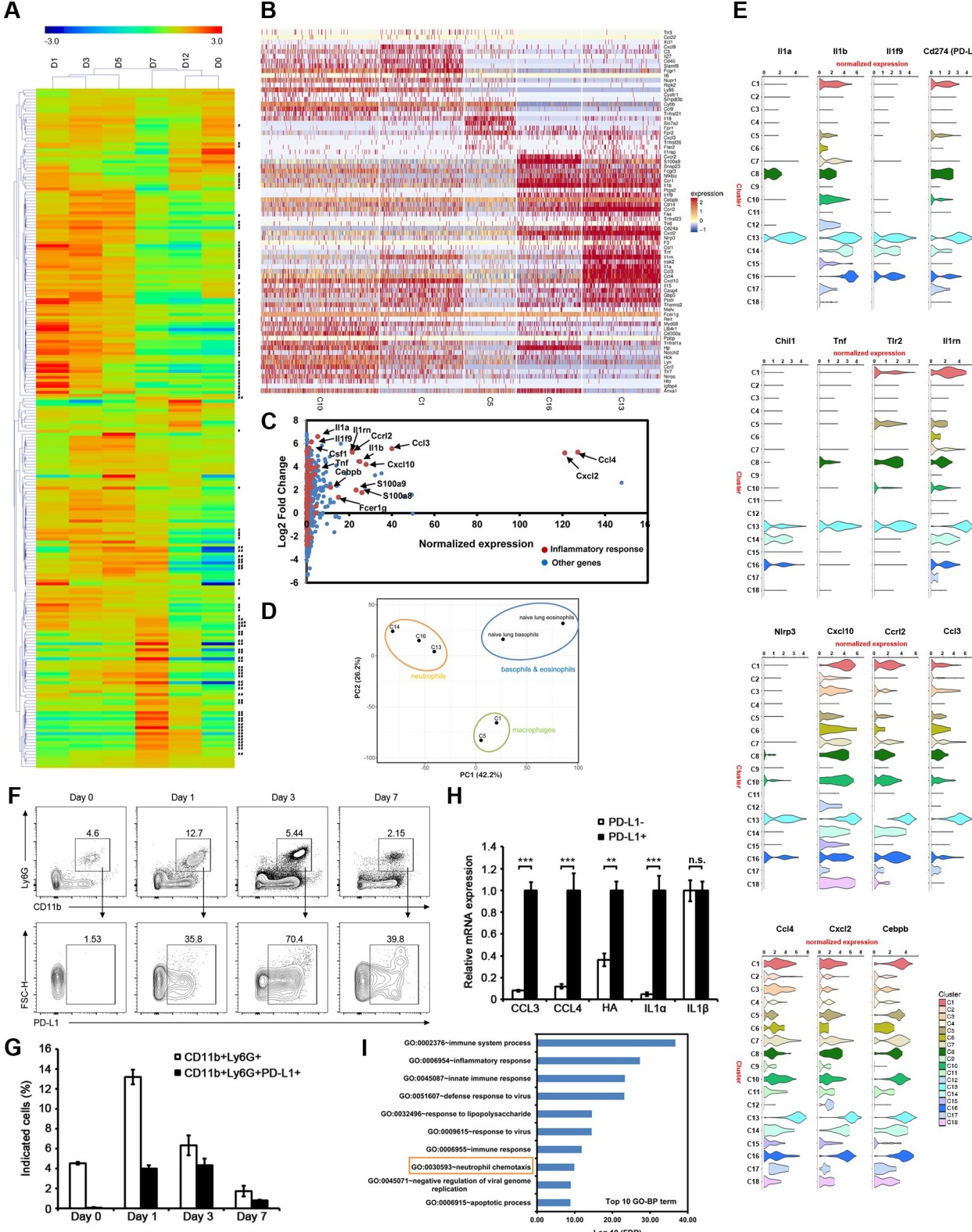

**Fig 2. A group of IAV-infected PD-L1+ neutrophils is the major contributor to the first wave of pro-inflammatory factors.** (A) Heatmaps showing the time-course log2 fold change of the pro-inflammatory genes (GO: 0006954). The right rows marked with # indicated the highly-expressed genes (Log2 fold change ≥1) at day 1 p.i.. The right rows marked with two ## indicated the highly-expressed genes (Log2 fold change ≥1)

at day 7 p.i. The hierarchical cluster trees were constructed based on the distance metric of pearson correlations among genes or libraries of different days post infection. (**B**) Heatmaps showing the normalized expression (Z-score) of the pro-inflammatory genes in various cells of clusters with high ratios at day 1 p.i.. The z-score normalization method was used to normalize the UMI counts of the single-cell transcripts. (**C**) Scatterplot showing the normalized expression profile of the cells in C13. Red dots indicated the pro-inflammatory genes, most of which were upregulated. (**D**) PCA plot showing the relationships among cell clusters together with the eosinophils and basophils from naïve the lung described by others (SRP040656, NCBI). X and Y axis show principal component 1 and principal component 2 that explain 42.2% and 26.2% of the total variance, respectively. (**E**) The normalized expression (UMI counts) of some significant genes with high expression including IL1α and IL1β in different clusters at day 1 p.i.. (**F**) Isolation of C13-neutrophils. Cells of the lung from mice infected with IAV at the indicated times post infection or from uninfected mice were collected. The Ly6G- and CD11b double positive cells were isolated (top panel), followed by staining with anti-PD-L1 (bottom panel). (**G**) Frequency of CD11b$^+$Ly6G$^+$ cells or CD11b$^+$Ly6G$^+$PD-L1$^+$ cells among total cells of the lung in F (Day 0 $n$ = 3, Day 1 $n$ = 5, Day 3 $n$ = 4, Day 7 $n$ = 4). Data are representative of three independent experiments. (**H**) The relative mRNA expression of pro-inflammatory genes in PD-L1$^+$ neutrophils (CD11b$^+$Ly6G$^+$PD-L1$^+$) and PD-L1$^-$ neutrophils (CD11b$^+$Ly6G$^+$PD-L1$^-$) from the lung of mice infected with 0.5 LD$_{50}$ of influenza A/ PR/8/34 (H1N1) viruses was analyzed with qRT-PCR at day 2 p.i.. Data are shown as the means ± SD in one of three independent experiments. $^{**}$, $P < 0.01$; $^{***}$, $P < 0.001$ (Student $t$ test, $n$ = 3). n.s. means not significant. (**I**) The top ten terms of Gene Ontology (GO) biological processes in the significant enrichment of the highly expressed genes in C13-neutrophils. Transformed false discovery rate (FDR) was indicated at the X-axis. The GO term of neutrophil chemotaxis was highlighted with rectangle.

including high levels of Ccl3, Ccl4, Cxcl2, Cxcl10, TNF-α, and IL1α at day 1 p.i. (Fig 2B and 2C). The heatmap constructed based on the log2 fold change of these pro-inflammatory genes in various cell clusters with high ratios at day 1 p.i. indicated that most pro-inflammatory factors upregulated at day 1 p.i. were enriched in C13-cells (S9 Fig). The graphical representation with tSNE plot of some classical pro-inflammatory genes which were highly-expressed in the single cell library at day 1 p.i.. were also enriched in the C13-cells (S10 Fig). The UMI counts of these pro-inflammatory gene transcripts at single-cell resolution also showed the highest values in C13-cells at day 1 p.i. compared with that in other clusters (S11 and S12 Figs). Pearson correlation analyses (PCA) for the patterns of antiviral factors and inflammatory response further demonstrated that C13 cells showed characteristics of granulocytes-like rather than that of macrophage/monocytes (S13 Fig). Moreover, PCA analysis confirmed that the character of C13 cells was significantly different from that of eosinophils and basophils whose sequencing data were deposited in the NCBI Sequence Read Archive database under the accession code SRP040656), but much closer to that of C14 and C16 which were typical neutrophils (Fig 2D). The cell transition among C14-C16-C13 was further demonstrated by constructing the single-cell trajectories in pseudotime (S14 Fig). Based on the highly expressed pro-inflammatory factors, the C13 cells were activated to a higher degree than C14 and C16 cells. As PD-L1 (CD274) was highly and specifically expressed in C13-neutrophils (Fig 2E), we chose PD-L1 as a marker for isolating C13 cells. Data from flow cytometry showed that PD-L1$^+$ neutrophils accounted for 30–70% of the total CD11b$^+$Ly6G$^+$ neutrophils in the early stage of IAV infection in the lung (Fig 2F and 2G). The isolated PD-L1$^+$ neutrophils (PD-L1$^+$CD11b$^+$Ly6G$^+$) at day 2 p.i. harbored much higher viral RNA and IL-1α mRNA levels, as well as higher Ccl3, Ccl4, IL-1β mRNA levels, compared with PD-L1$^-$ neutrophils (PD-L1$^-$CD11b$^+$Ly6G$^+$), indicating that the C13 PD-L1$^+$ neutrophils generated high pro-inflammatory cytokine mRNA level in the lung at early stage of IAV infection (Fig 2H and S15A Fig). The high mRNA and protein level of virus HA was also detected in C13 PD-L1$^+$ neutrophils (S5 Table and S15B Fig). Furthermore, gene ontology (GO) enrichment analysis showed that the primary function of C13-granulocytes were related to pro-inflammatory responses and neutrophil chemotaxis (false discovery rate (FDR) <1E-10) (Fig 2I). Therefore, we identified that a group of PD-L1$^+$ neutrophils were the major contributor to the first wave of pro-inflammatory factors at day 1–3 p.i.. However, it is notable that C18-epithelial cells, which have been considered as the major target for IAV infection, counted for only 1% in total cells of lung in our preparation. Since the lung were homogenized using lung dissociation kit rather than enzymatic digestion, the adhesive epithelial cells, especially highly infected epithelial cells which undergo apoptosis, would be largely removed after filtered through a 70 μm nylon mesh filter. Therefore, our data cannot exclude

the possibility that epithelial cells could also release a significant amount of pro-inflammatory factors at the earlier time.

## Pf4+-macrophages are the major contributor to the second wave of pro-inflammatory factors

To identify the cell types involved in generating the second wave of pro-inflammatory factors, we compared the transcriptional profiles of inflammatory responses among the major cell types at day 7 p.i.. Although C6-active T-cells expressed some pro-inflammatory factors, such as Ccl5, most pro-inflammatory factors such as Ccl7, Ccl8, Cxcl2, Ccl2, Ccl9, Ccl12, Cxcl10 (Fig 3A and S16–S18 Figs), TNF-α and complement family member C1q were mainly expressed in cells from C8 cells, which were characteristically Pf4-positive (Fig 3A and 3B). The heatmap constructed based on the log2 fold change of these pro-inflammatory genes in various cell clusters with high ratios at day 7 p.i. indicated that most pro-inflammatory factors upregulated at day 7 p.i. were enriched in the expression of C8-cells (S16 Fig). The graphical representation with tSNE plot of some classical pro-inflammatory genes which were highly-expressed in the single cell library at day 7 p.i.. were also enriched in the C8-cells (S19 Fig). The UMI counts of these pro-inflammatory gene transcripts at single-cell resolution also showed the highest values in C8-cells at day 7 p.i. compared with other clusters (S19 and S20 Figs). The Pf4+ cells were significantly increased at day 7 p.i. in the lungs of mice infected with IAV (Fig 3C and 3D). We confirmed that the pro-inflammatory factors Ccl2 and Ccl8 were expressed in these Pf4+ cells in the lung at day 7 p.i using an immunohistochemistry assay (Fig 3E and S21 Fig). As Pf4-positive megakaryocytes were recently identified in lung [29, 30], we compared the transcriptional profile of C8 with that of C5, C1, and C17, which were closed to C8 in tSNE maps, as well as that of megakaryocytes identified in lung and bone marrow [29]. We found that the C8 was much closer to C5/C1/C17-macrophage-lineage in the pearson cor-relation analysis (Fig 3F) and PCA analyses on the z-score normalized mean expression pro-files of these cell clusters (Fig 3G). We therefore proposed that these C8 Pf4-postive cells were macrophage-lineage rather than megakaryocyte-lineage. Accordingly, we found that a group of Pf4+-macrophages was the major contributor to the second wave of pro-inflammatory fac-tors at day 7 p.i..

To further decipher the function of C8-Pf4-positive macrophages in the lung after IAV infection, we utilized a Pf4-cre induced DTR line (*Pf4-cre; iDTR*) in our study [31]. Pf4+CD41+-macrophages cells are depleted in the presence of diphtheria toxin (DT). The *Pf4-cre; iDTR* mice were intraperitoneally injected daily with DT 4 days after inoculation with 0.5 LD50 of A/PR/8/34 (H1N1) viruses (Fig 4A). Of note, the *Pf4-cre+; iDTR* mice and *Pf4-cre-; iDTR* mice were injected with DT after infected with A/PR/8/34 (H1N1) viruses. The cell num-bers of neutrophils and other macrophages were unaffected in the DT system (S22A Fig), implying that the Pf4+ macrophages were specifically depleted in *Pf4-cre+; iDTR* mice. Of note, Pf4+ macrophages belong to CD11b+Ly6C+ macrophages (S22B Fig). To examine whether the depletion of C8 Pf4-positive macrophages would affect the secretion of second-wave of pro-inflammatory factors *in vivo*, we analyzed the expression of Ccl7, Ccl8, Ccl12, Cxcl12, Spp1, and Cxcl3 with RT-PCR analysis (Fig 4B), and the expression of Ccl8 and Ccl2 with ELISA assays (Fig 4C and 4D) at day 8 p.i.. We found that the expression of pro-inflammatory factors was decreased in *Pf4-cre+; iDTR* mice when compared with that in *Pf4-cre-; iDTR* mice, further indicating that C8 Pf4-positive macrophages were the major contributor to the second wave of pro-inflammatory factors.

Besides releasing a wave of pro-inflammatory factors, we found that C8 Pf4-positive macro-phages had high expression of Pparg (Fig 4E), a signature transcription factor that is essential

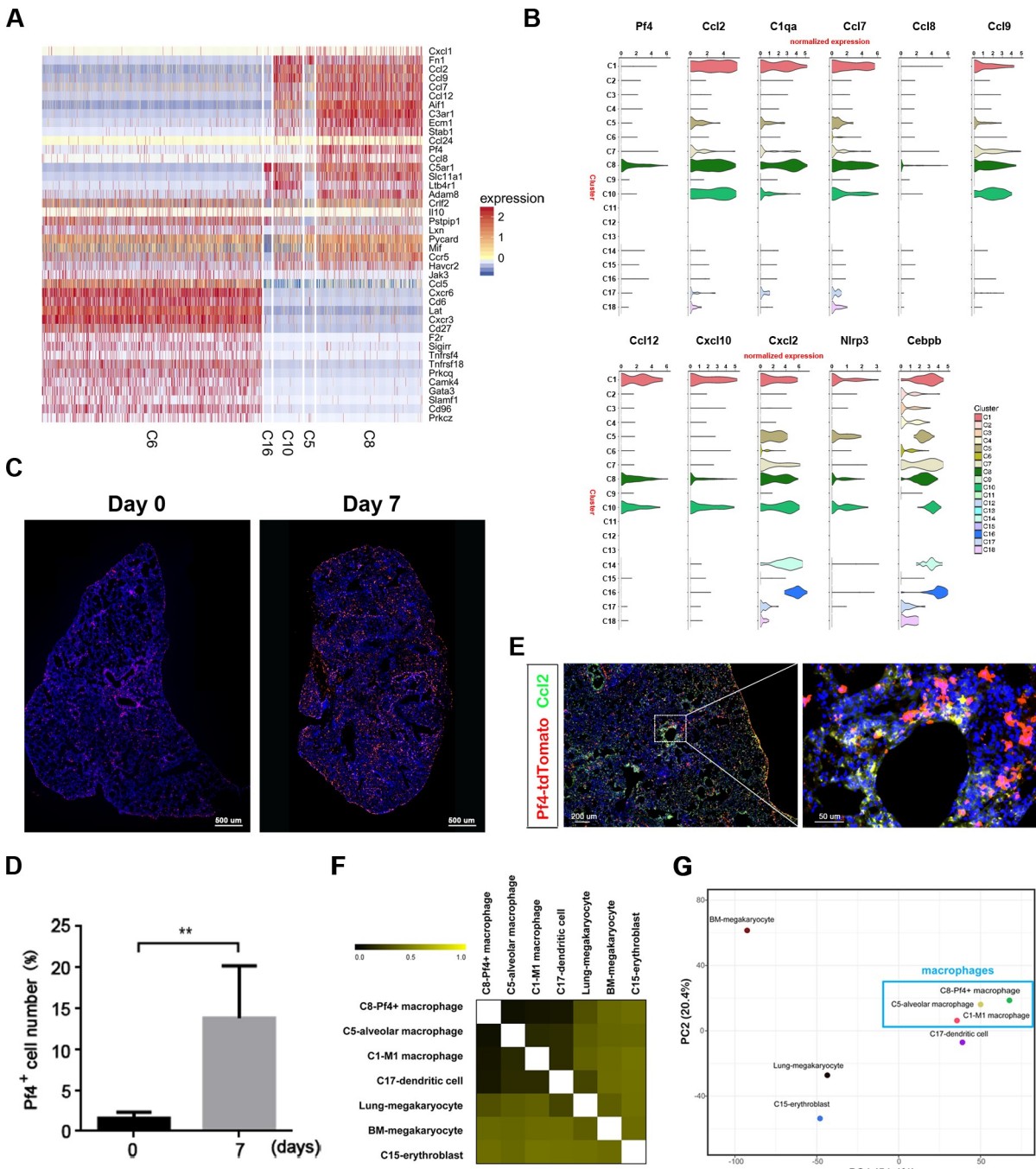

**Fig 3. Pf4+-macrophages are the major resource to generate second wave of pro-inflammatory factors at late stage of IAV-driven pneumonia.** (**A**) Heatmaps showing the normalized expression (Z-score) of the pro-inflammatory genes in the various cells of clusters with high ratios at day 7 p.i.. (**B**) The normalized expression (UMI counts) of some significant genes with high expression in different clusters at day 7 p.i.. (**C**) Pf4+ cells in the lung after IAV infection. The tdtomato-Pf4 mice were uninfected or infected with 0.5 $LD_{50}$ of influenza A/PR/8/34 (H1N1) viruses. At day 7 p.i, the tdtomato-Pf4+ cells in the lung were scanned. The nucleus was stained with DAPI (blue) and tomato-Pf4 (red) was shown in red. Scale bars, 500 μm. (**D**) At day 0 and day 7 p.i., the tdtomato-Pf4+ cells in the lung were calculated. At least 300 cells in each group from three independent assays were scored. Data are shown as the means ± SD. **, $P < 0.01$ (Student $t$ test, $n = 3$). (**E**) The expression of Ccl2 in Pf4+ cells. The tdTomato-Pf4 mice were infected with 0.5 $LD_{50}$ of influenza A/PR/8/34 (H1N1) viruses. At day 7 p.i, the tdTomato-Pf4+ cells in the lung were stained with anti-Ccl2 antibodies (green). The nucleus was stained with DAPI (blue) and Pf4-tdTomato (red) was shown in red. Scale bars, left, 200 μm; right, 50 μm. (**F**) Heatmap showing the scaled distances calculated based on pearson correlations for relationships between the z-score normalized mean expression profiles in all the indicated cells. (**G**) PCA plot showing the relationships among the indicated cell clusters and the megakarycytes from lung and bone mellow described by others (SRP097794, NCBI).

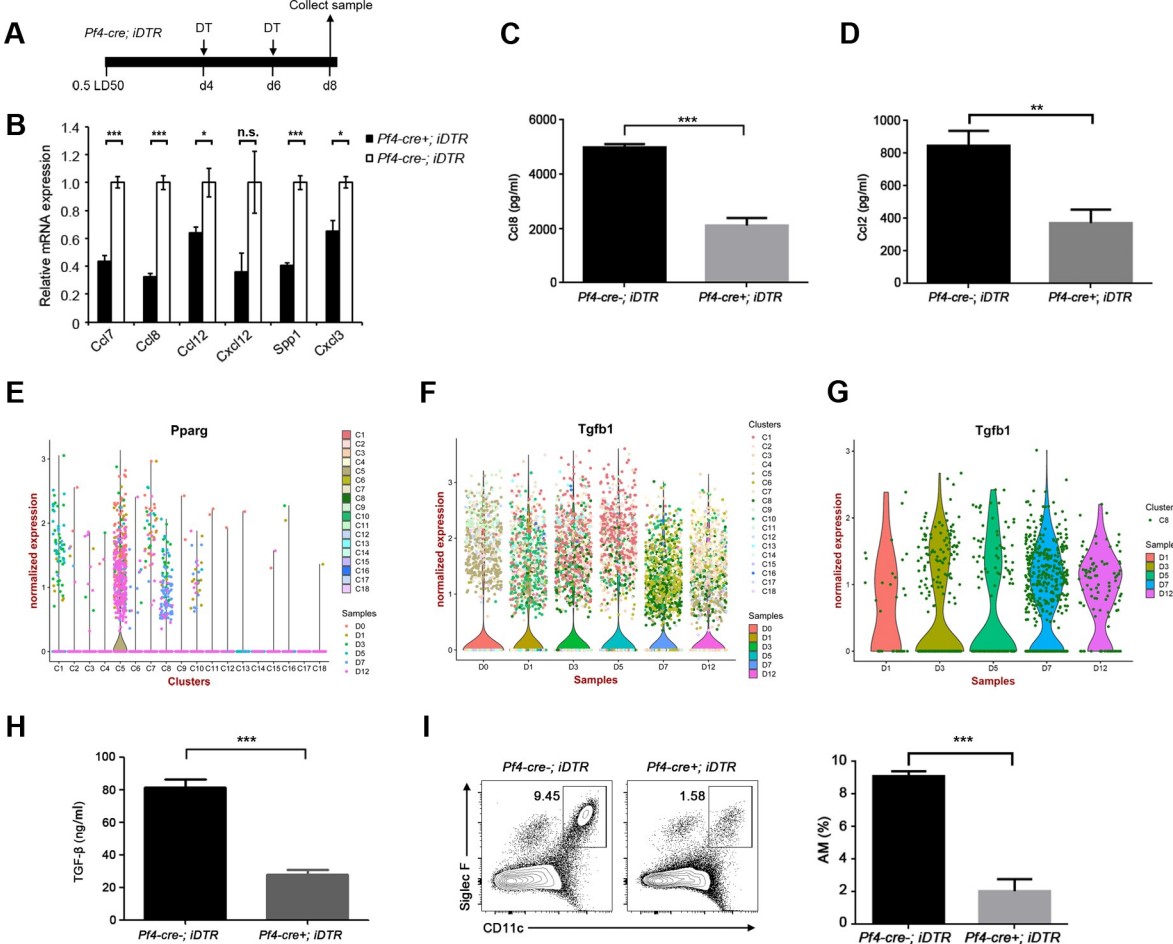

**Fig 4. Pf4+-macrophages are the precursors of AMs.** (**A**) Scheme for DT administration to *Pf4-cre; iDTR* mice after infection with 0.5 LD$_{50}$ of A/PR/8/34 (H1N1) virus used for the experiments shown in B–I. (**B**) The relative mRNA expression of pro-inflammatory genes in cells of the lung samples from *Pf4-cre⁻; iDTR* mice and *Pf4-cre⁺; iDTR* mice was analyzed with qRT-PCR at day 7 p.i.. Data are shown as the means ± SD in one of three independent experiments. *, $P < 0.05$; ***, $P < 0.001$ (Student $t$ test, $n = 3$). n.s. means not significant. (**C** and **D**) The Ccl8 (**C**) and Ccl2 (**D**) expression of BALs of the lung samples from *Pf4-cre⁻; iDTR* mice and *Pf4-cre⁺; iDTR* mice were analyzed with ELISA analysis. Data are shown as the means ± SD in one of three independent experiments. ***, $P < 0.001$ (Student $t$ test, $n = 3$). (E) The normalized expression (UMI counts) of Pparg in the lung with high expression in different clusters from different time point p.i.. (**F**) The normalized expression (UMI counts) of Tgfb1 with high expression in different clusters at day 7 p.i.. (**G**) The normalized expression (UMI counts) of Tgfb1 with high expression in C8 cluster at day 7 p.i.. (**H**) The TGF-β expression of the lung samples from *Pf4-cre⁻; iDTR* mice and *Pf4-cre⁺; iDTR* mice was analyzed with Elisa analysis. Data are shown as the means ± SD in one of three independent experiments. ***, $P < 0.001$ (Student $t$ test, $n = 3$). (**I**) Left panel: Flow cytometry of AMs (CD11c⁺Siglec F⁺) of the lung samples collected at day 8 p.i. from *Pf4-cre⁻; iDTR* mice and *Pf4-cre⁺; iDTR* mice. Numbers in quadrants indicate percent AMs cells. Right panel: Frequency of AMs (CD11c⁺SiglecF⁺) of the lung samples. Data are shown as the means ± SD in one of three independent experiments. ***, $P < 0.001$ (Student $t$ test, $n = 3$).

for the development of AMs. TGF-β (encoded by Tgfb1) was also highly expressed in C8 Pf4-positive macrophages (Fig 4F and 4G), which were the main cytokines for the development of monocyte-derived AMs. Also, the secretion of TGF-β was reduced after the depletion of C8 Pf4-positive macrophages, indicating that C8 Pf4-positive macrophages were the major contributors for the secretion of TGF-β (Fig 4H). Thus, we supposed that C8 Pf4-positive macrophages are the precursors of AMs during IAV infection. To this end, we analyzed the percentage of AMs at day 8 p.i. and found that AMs was significantly decreased after the depletion of C8 Pf4-positive macrophages (Fig 4I), while the percentage of T lymphocytes was

unaffected (S22C Fig). These results indicated that Pf4-positive macrophages may be the precursors of AMs, and contributed to the release of the second wave of pro- inflammatory factors.

## Identification the ligand/receptor pair among C13 and C8 and others clusters

In order to comprehensively elucidate the regulatory network of various infiltrated inflammatory cells in the lung during viral infection, we systematically analyzed the receptors and ligands of chemokines, interleukins, interferons and other inflammatory factors (Fig 5, S23 and S24 Figs). Highly expression of the inflammatory factors from IL-1 family such as IL-1α, IL1-β, and IL-RN was found in C13-PD-L1+ (CD274+) neutrophils at the early stage of infection (day 1 p.i.) (Fig 5A). Correspondingly, IL1R2 and IL1RAP were significantly upregulated in neutrophils cluster C13 and C16 and C10-monocytes, but not IL1R1 which are the receptors of these chemokines. Other chemokines like Ccl3 and Ccl4 were also highly expressed in C13, and the major receptors such as Ccr1 were found abundant in C13, C16, C10, C5-alveolar macrophage and C1-M1-macrophage. Moreover, the Cxcl2/Cxcr2 axis which regulates NLRP3 inflammasome activation was found in C13 and C16. To study the receptors of C13, we systematically summarized the receptors of C13 in day 1 p.i. (Fig 5B). As the major highly expressed receptors, IL1R2 and Ccr1 were found in these clusters, and the corresponding ligands were also highly expressed in C13 and C16, which were consistent with the result of ligands analysis (Fig 5A). In addition, C1 (M1-macrophage), C2, and C3 (NK cells) also contributed substantial ligands of Ccr1 such as Ccl2, Ccl3, Ccl4, and Ccl5.

C8 cluster secreted various cytokines at day 7 p.i when viruses were almost undetectable. High levels of Ccl2/Ccr2 and Ccr5 pair were exited in all the clusters including C8 cluster (Fig 5C and 5D). These pair bondings were also observed in Ccl3, Ccl4, Ccl5/Ccr1, and Ccr5. New chemokines such as Ccl7 and Ccl8 emerged in C8 at day 7 p.i.. The known receptors Ccr1, Ccr2 and Ccr5 were also highly expressed in almost all clusters. Importantly, we observed strong interaction between C8 and other cell clusters such as C5-AM clusters. Thus, through the ligand/receptor interacting map of lung during IAV infection, we identified various interlineage crosstalks and further confirmed the correlation and relationship between C8 and other cell clusters in convalescence of IAV infected in the lung.

## Discussion

The involvement of immune cells during IAV infection in the lung is a dynamic and complex process. To provide an atlas of immune response in the lung during IAV infection, we performed a scRNA-seq analysis of the transcriptional profile database of pulmonary immune cells during IAV infection. Previous reports have unveiled the novel antiviral factors in the early stage of IAV infection using single-cell analysis through comparing the infected and uninfected samples [23]. However, the exact profile of the IAV-driven immunopathogenesis in the lung is still unclear. Using the transverse (cells to cells) and longitudinal (day to day) analysis of immune cells in the lung, we gained insights into of the pulmonary immune processes during IAV infection.

Considering the strong connection between the severity of IAV infection and cytokine/chemokine production, the pro-inflammatory factors releasing cells were further analyzed [32]. By sequentially analyzing the transcriptome of infiltrated immune cells, we identified two waves of pro-inflammatory factor released. C13, C10, and C16 contributed the most to early pro-inflammatory factor release (Day 1 and Day 3). In particular, the IAV-infected C13-PD-L1+-neutrophils are the major contributor to the first wave of pro-inflammatory

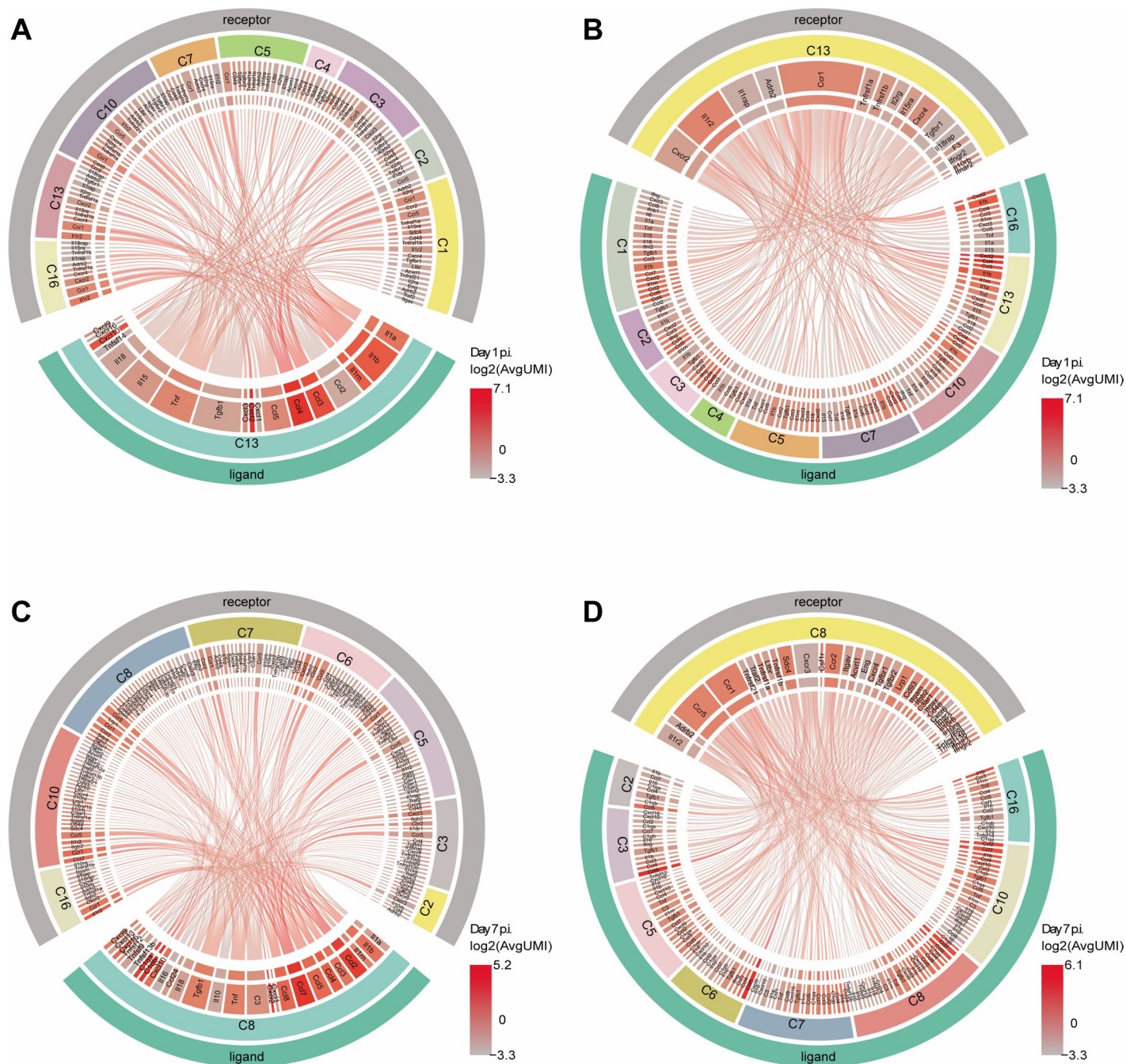

**Fig 5. Identification the ligand/receptor pair among C13 and C8.** (**A-D**) Circos plots showing the ligand and receptor interactions of the immune cells in C13 or C8 of lung at day 1 (**A** and **B**) or day 7 p.i. (**C** and **D**). Line connections indicate the literature supported ligand and receptor interactions. The average UMI counts of all LR genes were log2 transformed, and highlighted with the gradual red color in the graph according the transcription level of LR genes. The LR interaction lines were colored in accordance with the transcription level of ligand or receptor genes in the main cluster.

factor release. Signaling through IL-1R contributed to both host protection and immuno-pathogenesis following IAV infection as previously reported [33]. In our study, high expression of both IL-1α and IL-1β was detected in C13/C16-neutrophils, while IL-1α was only generated by C13-neutrophils. The C13-neutrophils may be recruited from bone marrow or

transformed *in situ* in the lung, which remains to be further explored. The high level of virus infection in C13-neutrophils was revealed by the single-cell sequence and qRT-PCR. It is notable that the immune reaction and virus replication were so extensive in this cell cluster simultaneously. NS of IAV is the most important viral protein that counteracts the IFN system, and we found that the level of NS RNA was particularly high in C13-neutrophils when compared with other clusters. The balance between immune reaction and immune antagonism in this cluster during IAV-driven immunopathogenesis reminds to be clarified.

Notably, after IAV copy is hardly found in lung, the second wave of pro-inflammatory factors was generated, mainly by a group of special cells, which was identified as Pf4$^+$-macrophages. C8-Pf4$^+$-macrophages expanded up to 18% of total cells at day 7 p.i.. This cluster expresses high level of cytokines and chemokines such as Ccl7, Ccl8, Cxcl2, Ccl2, Ccl6, Ccl9, Ccl12, Cxcl10, TNF-α, Trem2, and complement family member C1q at day 7 p.i.. The depletion of Pf4$^+$-macrophages led to a reduction in cytokines such as Ccl2, Ccl7, and Ccl8, which were reported to recruit monocytes and facilitate the maturation of tissue-specific macrophages. Besides these pro-inflammatory factors such as TNF-α directly worsen the immune injury, C8-Pf4$^+$-macrophages may assist the recruitments of monocytes or T-lymphocytes in lung through binding to Ccr2 by generating Ccl2 and Ccl7. Furthermore, high level expression of C1qa, C1qb, and C1qc was also detected in C8-Pf4$^+$-macrophages. As a recent study reported that C1q could regulate the activation of CD8$^+$ T cells in autoimmunity and viral infection, it is possible that C8-Pf4$^+$-macrophages may assist T lymphocytes in adaptive immunity [34]. Collectively, we especially identified the origin of cytokine/chemokine occurring at late stage of immunopathogenesis, which could contribute to the persistence of severe IAV-driven pneumonia.

Interestingly, C8-Pf4$^+$-macrophages also expresses high level of TGF-β at day 7 p.i., which were reported to promote the mature of AMs in the lung. Considering that the PPAR-γ was highly expressed in C8-Pf4$^+$-macrophages, and the depletion of Pf4$^+$-macrophages led to a significant reduction in AMs formation in the lung at late stage IAV-mediated pneumonia, we hypothesized that C8 Pf4$^+$-macrophages may be the precursors of AMs when lung recovered from IAV infection. The characteristic of monocyte-derived AMs was different from tissue resident AMs. These cells may transform to AMs along with a lot of cytokine release, and the autosecreting of TGF-β in C8 Pf4$^+$-macrophages further promoted these cells to differentiate into AMs. Collectively, we especially identified the origins of cytokine/chemokine release occurring at the late stage of immunopathogenesis, which were probably the precursors of AMs.

Inflammatory factors play important roles in the infectious diseases. Studying the communication networks between immune cells can help us understand infectious diseases deeply. Reducing immune response in cytokine storm and enhancing the immune response to eliminate pathogens efficiently were the aim for IAV-induced pneumonia treatment. Through comprehensive analysis of ligands and receptors interaction of infiltrating immune cells in lung, we found strong autocrine loop in immune cells both in early and convalescent stages of infection. For instance, the ligands (IL-1α, IL1-β, and IL-RN) and receptors (IL1R2 and IL1RAP) of IL-1 system were both highly expressed in C13 neutrophils cluster compared with other clusters. In addition, the upregulation of Ccl3 and Ccl4 was also accompanied by the high levels of Ccr1, and Cxcl2/Cxcr2 axis, which was also remarkable in C13 and C16. Controlling these autocrine loops to avoid the excessive release of inflammatory factors was critical for preventing the production of cytokine storm. IAV infection induced pneumonia would cause plenty of epithelial cells apoptosis and inflammatory lung injury. When the virus elimination at day 7 p.i., monocyte/macrophage massively infiltrated into lung. Moderate of monocyte/macrophage and inflammatory factors favors lung damage repair. However, uncontrollable cell

invasion and cytokine released would induce serious pulmonary immune pathology or pulmonary fibrosis. The mechanism that C8 Pf4$^+$ monocyte/macrophage released so strong cytokine remained to be further explored. For example, how cytokines such as Ccl2, Ccl3, Ccl4, Ccl7, and Ccl8 transduce different signal even in a single cell still unknown.

In summary, through scRNA-seq analysis, we demonstrated many new phenomena for the pathogenesis of IAV infection. Importantly, by sequentially analyzing the transcriptome of infiltrated immune cells, we identified two waves of pro-inflammatory factor releases mainly from two cell clusters, which have not been described previously. These clusters could be the major origin of cytokine/chemokine storm occurring in IAV-driven pneumonia [5, 35]. Therefore, these newly-identified clusters should be considered as important therapeutic targets for IAV-driven pneumonia, for relieving the clinic symptoms caused by cytokine storm, for shortening the time for recovery, or for preventing severe sequential complications.

## Methods

### Mice

C57BL/6, Pf4-cre, *iDTR*, and Rosa26-LSL-tdTomato mice on a C57BL/6 background were used in our study. To track Pf4$^+$ cells, Pf4-cre mice were crossed with Roso26-LSL-tdTomato mice to generate Pf4-tdTomato-expressing cells. To generate *Pf4-cre; iDTR* mice, Pf4-cre mice were crossed with the *iDTR* line. To induce *Pf4-cre; iDTR* mice model, DT was injected intraperitoneally twice at day 4 and day 6 p.i. at a dose of 50 ng/g. Pf4-cre and Rosa26-LSL-tdTomato mice were provided by Dr. Linheng Li lab of Kansas University. Mice aged 6–12 weeks were used at the start of the experiments. Littermate controls were used in all experiments. Mice were bred and housed in specific pathogen-free conditions at the Animal Center of Sun Yat-sen University (SYSU) Zhongshan School of Medicine (ZSSOM) *via* standard pellet feed and water.

### Ethics statement

All animal experiments were carried out in strict accordance with the guidelines and regulations of Laboratory Monitoring Committee of Guangdong Province of China, and were approved by Ethics Committee of Zhongshan School of Medicine (ZSSOM) of Sun Yat-sen University on Laboratory Animal Care (Assurance Number: 2017–061).

### Influenza infection

Mice were anaesthetized with isoflurane and inoculated intranasally with 0.5 LD$_{50}$ (50 PFU) of influenza A/PR/8/34 (H1N1) viruses. The lungs were collected at the indicated time points post infection (p.i.). The virus stocks were obtained from embryonated chicken eggs after inoculation for 48–72 h, and the titers were determined by a plaque assay on MDCK cells, as described previously [36]. Body weight and survival rates of each group were measured daily.

### FACS analysis and cell sorting

Single cell suspensions were prepared from the lung, spleen, bone marrow or blood samples. Cells were incubated with anti-Fc receptor antibodies (clone 2.4G2) and stained with the antibodies on ice for 20 min before washing. For intracellular staining, cells were stained with antibodies to surface molecules, followed by being fixed and permeated in Cytofix/Cytoperm buffer (BD Biosciences). Cells were stained intracellularly and then analyzed using a LSRFortessa (BD Biosciences) or sorted with a FACSAria (BD Biosciences) following the manufacturer's procedures. Data were analyzed with FlowJo software (TreeStar).

All monoclonal antibodies used for flow cytometry were from eBioscience, unless stated otherwise: anti-mouse CD11b (M1/70), anti-mouse SiglecF (1RNM44N), anti-mouse Ly6C (ER-MP20), anti-mouse Ly6G (1A8-Ly6g), and anti-mouse CD11c (N418).

## Immunofluorescence assay (IFAs)

Cells were fixed with 4% paraformaldehyde (PFA), and permeabilized with 1% Triton X-100, followed by blocking with 5% Goat Serum in PBS blocking solution and stained with primary antibodies for 1h. After being washed for 3 times, cells were stained with anti-rat Alexa-Fluor488 and anti-rabbit AlexaFluor594 secondary antibodies for 1 h, followed by washing for 3 times again and staining with 4', 6-diamidino-2-phenylindole (DAPI) reagent (Invitrogen). All the procedures were performed at room temperature.

## Quantitative real-time PCR (qRT-PCR)

Erythroblasts, granulocytes, or M1 macrophages were isolated and the total RNA of each cell was extracted for qRT-PCR. cDNA was reversed with oligo(dT) and random hexamers using the PrimeScript RT reagent kit (Takara). Real-time PCR was performed using SYBR Green (Bio-Rad) with a qTOWER2.0 (Analytik Jena AG). Relative expression was determined by normalization to the housekeeping gene β-actin with Bio-Rad CFX Manager software.

## Immunohistochemistry (IHC)

The lungs were fixed in 2% paraformaldehyde for 24 h and embedded in paraffin. After freezing, the paraffin blocks were sectioned into 5 μm slides, adhered onto the glass slides, and fixed in ice-cold acetone. Sections were pretreated with Image-iT FX Signal Enhancer (Thermo Fisher Scientific) and blocked with with 5% Goat Serum in PBS blocking solution. Sections were then stained with anti-Ccl2 (Rabbit, Bioss) primary antibodies, followed by staining with anti-rat AlexaFluor488 and anti-rabbit AlexaFluor594 secondary antibodies, and DAPI.

## ELISA assay

Mouse Ccl2 and Ccl8 ELISA kit was purchased from R&D. The bronchoalveolar lavage (BAL) of the lung samples from *Pf4-cre⁻; iDTR* mice and *Pf4-cre⁺; iDTR* mice was harvested from 96 wells plate and then detected the Ccl2 or Ccl8 concentrations by following the instruction of manufacturer.

## Single cell RNA-seq

**(1) Single cell collection and cDNA amplification.** Single cell capture was performed using a Chromium Controller instrument (10x Genomics), a highly repeatable, efficient and stable solution for cell characterization and gene expression profiling of thousands to millions of cells (https://www.10xgenomics.com/solutions/single-cell/). Single cells were collected from the lungs of mice (three mice per group) uninfected (day 0) or infected with A/PR/8/34 (H1N1) virus at 5 time points including day 1, day 3, day 5, day 7, and day 12 p.i.. The lung tissue was dissected and homogenized using a lung dissociation kit (Miltenyi Biotec). Following dissociations, single cell suspensiosn were filtered through a 70 μm nylon mesh filter (BD Biosciences) into PBS supplemented with 0.2 mM pH8 EDTA and 0.04% bovine serum albumin (BSA). Red blood cells were lysed by hypotonic lysis. Fresh cells from the lung were harvested, washed with 1× PBS and re-suspended at 1× 10⁶ cells per ml in 1× PBS containing 0.04% BSA to minimize cell loss and aggregation following the protocol recommended by 10x Genomics. Cell viability of the samples was analyzed using trypan blue exclusion staining to ensure more

than 90% of live cells. Cellular suspensions were loaded on the Chromium Controller instrument to generate single-cell gel bead-in-emulsions (GEMs) with Chromium single cell 3' reagent v2 kits (10x Genomics), containing a pool of ~750.000 barcodes sampled to separately index the transcriptome of each cell. Thousands of individual cells were isolated into droplets together with gel beads coated with unique primers bearing 10X cell barcodes, unique molecular identifiers (UMI) and poly (dT) sequences. According to the single cell 3' reagent kit protocol, GEM-reverse transcriptions were performed in a Veriti 96-well thermal cycler (Thermo Fisher Scientific). After RT, GEMs were broken and the barcoded single-strand cDNA was cleaned up with DynaBeads MyOne Silane Beads (Thermo Fisher Scientific) and a SPRI select Reagent Kit (Beckman Coulter). Global Amplification of cDNA was achieved using the Veriti 96-well thermal cycler, and the amplified cDNA product was cleaned up with the SPRIselect Reagent Kit.

**(2) Library construction and sequencing.**   The indexed sequencing libraries were constructed using the reagents in the Chromium Single Cell 3' Library v2 Kit for (a) fragmentation, end repair and A-tailing; (b) size selection with SPRI select beads; (c) adaptor ligation; (d) post-ligation cleanup with SPRI select beads; and (e) sample index PCR and final cleanup with SPRI select beads. The final single cell 3' library comprises the standard Illumina paired-end constructs which begin and end with P5 and P7 primers used in Illumina bridge amplification. The barcoded sequencing libraries were quantified by a Bioanalyzer Agilent 2100 System using a High Sensitivity DNA chip (Agilent), and the quantitative PCR using a KAPA Library Quantification Kit (KAPA Biosystems). Finally, the sequencing libraries were loaded onto a HiSeq2500 (Illumina) with a custom paired-end sequencing mode (26 bp for read 1 and 98 bp for read 2) to obtain a sequencing depth of ~50,000 reads per cell.

## scRNA-seq bioinformatics analysis

**(1) Initial quality control.**   The single-cell sequencing files (base call files) were processed using the Cell Ranger Single-Cell Software Suite (v2.0) for quality control, sample demultiplexing, barcode processing, and single-cell 3'gene counting [37]. The raw base call files of each sample were first demultiplexed into fastq data using the bcl2fastq conversion software. Quality control of the fastq data was performed using FastQC software, and the data were aligned to the Nucleotide Sequence Database (https://www.ncbi.nlm.nih.gov/genbank/) using the basic local alignment search tool (BLAST) to avoid the data distortion caused by the experimental contamination of other species, especially the bacterial infection or contamination. After the initial quality control, the sequences with barcodes and UMIs of low quality were removed. We obtained about 862.7 million clean reads based on the mouse transcriptomes of 16,424 cells, achieving >50,000 mean reads per cell. More than 98% of the clean reads had high quality scores at the Q30 (an error probability for base calling of 0.1%) level in the bases of the barcodes and UMIs. The sequencing saturation of each sample was above 80%, and 15,237~16,505 mouse genes were detected across six single-cell RNA-seq libraries.

**(2) Alignment, UMI counting and multi-library aggregation.**   The fastq data were aligned to the UCSC mouse reference genome (mm10) using STAR with default parameters. For further counting of the UMI tags, the CellRanger count algorithm was used to generate single-cell gene counts for a single library, which can provide the most stable and accurate clustering solutions for 10x Genomics scRNA-seq data [26]. Only confidently mapped, non-PCR duplicates with valid barcodes and UMIs were used to generate the gene-barcode matrix. For quantitatively identifying intracellular viral segmented mRNAs to track the cells from the lung infected with IAV at single-cell resolution, the scRNA-seq data of six lung samples of the lung from mice uninfected (day 0) or infected with A/PR/8/34 (PR8, H1N1) virus at 5 time

points including day 1, day 3, day 5, day 7, and day 12 were reanalyzed using the CellRanger count algorithm based the union of mm10 and PR8 (txid211044, NCBI) reference genome. For the comparison of the scRNA-seq data among different libraries, the gene-cell-barcode matrix of each sample was normalized by equalizing the read depth between libraries for further merging using the CellRanger aggregate procedure, which was confirmed using the Seurat integrated analysis method [38]. The reads from higher-depth libraries were subsampled until all libraries have an equal number of confidently mapped reads per cell. The gene-cell-barcode matrix from each of the six samples was concatenated, log-transformed and filtered based on the number of genes detected per cell. Any cell with less than 200 genes or more than 30% of mitochondrial UMI counts was filtered out, and only genes with at least one UMI count detected in at least one cell were used for further analysis, which was performed using CellRanger R version 2.0.0 and Seurat suite version 2.0.0.

(3) **Clustering, differential expression and visualization.** For clustering the cells, the principal component analysis (PCA) was run on the normalized filtered gene-barcode matrix to reduce the number of feature (gene) dimensions. Top 15 principal components (PCs) were selected and passed to t-distributed Stochastic Neighbor Embedding [38] for clustering visualization in a two dimensional space. Graph-based Clustering was then run to group cells together that have similar expression profiles, building a sparse nearest-neighbor graph without pre-specification of the number of clusters. Clusters were grouped into 18 unsupervised categories, according to the differential expression profile with hallmark genes. To identify genes that were enriched in a specific cluster, the mean expression of each gene was calculated across all cells in the cluster. Then each gene from the cluster was compared to the median expression of the same gene from cells in all other clusters, and the log2 fold-change of differentially expressed gene was calculated. For hierarchical clustering, pairwise Pearson correlation between each cluster was calculated based on the mean expression of each gene across all cells in the cluster, and the log2 fold-change of differentially expressed gene was used for visualization by heatmap with MEV software (http://www.tm4.org/). The graphical representation of specific gene expression with tSNE plot was implemented by using Loupe Cell Browser software and Cell Ranger R.

(4) **Single cell trajectory analysis, PCA analysis and Gene Ontology enrichment.** The single cell trajectory analysis was performed using the package Monocle for constructing single-cell trajectories in pseudotime based on the differential expressed genes among related single cells. The transcriptional profile data of macrophage, eosinophils and basophils were retrieved from the NIH SRA database with the accession code SRP040656 (https://www.ncbi.nlm.nih.gov/sra/). After the z-score normalization, the transcriptional profile data of macrophages, eosinophils and basophils from database together with the specific cell clusters in our study were used for pairwise Pearson correlation and PCA analysis implemented in R language (http://www.r-project.org) to demonstrate the phylogeny of specific clusters. Functional pathways representative of each gene signature were analyzed for enrichment in gene categories from the Gene Ontology Biological Processes (GO-BP) database (Gene Ontology Consortium) using DAVID Bioinformatics Resources [39].

## The ligand and receptor interaction maps

To visualize the ligand and receptor (LR) interactions of the immune cells in lung post of IAV infection, the published dataset of ligand and receptor pairs [24] were used as the reference. The low average UMI count of the LR genes were filtered, including all LR genes with <0.1 UMI counts average in each cluster (each sample was normalized by equalizing the read depth). We built an interaction circular graph by referring the literature supported ligand and

receptor pairs, and connecting the edges between them, generating with the Circos package (http://circos.ca/). We computed the log transformed UMI counts, and the highly expressed ligand and receptor interactions were highlighted with the red color in the graph using the Circos package.

## Statistical analysis

Data were analyzed using GraphPad Prism 6.0 software (La Jolla, CA, USA). The two-tailed Student's *t*-test was used to determine the significance of statistical data between two experimental groups or multiple comparisons. Data were considered significant at $^*P < 0.05$, $^{**}P < 0.01$ and $^{***}P < 0.001$.

## Supporting information

**S1 Table. The quality control of scRNA-seq.**
(TIF)

**S2 Table. Cell numbers of different cell clusters in each libraries at different days post-infection with IAV.**
(TIF)

**S3 Table. Pearson correlation for relationships between the normalized expression profiles of cells from different clusters.**
(TIF)

**S4 Table. The pearson correlation between the sample data form multiple libraries at different days post-infection.**
(TIF)

**S5 Table. The average UMI counts of IAV eight genes in different cell clusters.**
(TIF)

**S6 Table. The UMI counts of IAV eight genes in different cell clusters.**
(TIF)

**S1 Fig. Landscape of cell populations in the lung during IAV infection.** (A) tSNE projection where each cell is colored by log10 of UMI counts. Color scale represents log10 of UMI counts. Each point in the scatter plot represents a cell in the coordinates specified by the two t-SNE components. The color of each point plotted indicates the total number of UMIs for each cell, and these count values are displayed in log10 scale. (B) tSNE maps displaying 16,424 suspended cells from the lung and coloured by the main cell populations based on the unsupervised graph-based, showing the formation of 18 main clusters with the cell numbers in the right panel. (C) Heatmap showing the scaled distances calculated based on pearson correlations for relationships between the normalized mean expression profiles of cells from different clusters. A hierarchical cluster tree constructed based on the distance metric of pearson correlation was shown at the left panel. The numbers represent the percent of all cells from that cluster that are in each day's library. (D) Cells of different clusters in the lung were analyzed by FACS analysis at different day post infection. Cells of the lung from mice infected with IAV at the indicated times post infection or from uninfected mice were collected. C6-CD8$^+$ T cells, C8-Pf4$^+$-macrophages, and C13-PD-L1$^+$-neutrophils in the lung were analyzed with FACS analysis, and the cell numbers or frequency were calculated. Data are representative of three independent experiments.
(TIF)

**S2 Fig. Major cell categories and their signature genes.** (**A**) the signature genes of each cluster was shown. (**B**) PCA analysis of the 18 main clusters Graph-based Clustering. X and Y axis show the principal component 1 and principal component 2 that explain 48% and 16.3% of the total variance, respectively.
(TIF)

**S3 Fig. The determination of IAV-infected cells during IAV infection.** (**A**) tSNE maps displaying 16,424 cells from the lung of mouse after infected with IAV and colored by the samples of different days post-infection. tSNE maps of different days were combined for comparison with corresponding colors. The data between libraries was normalized by equalizing the read depth between libraries before merging. (**B**) tSNE maps displaying the comparisons between samples of different days post infection. (**C**) Proportions of different cell clusters in each library at different days p.i..
(TIF)

**S4 Fig. Significantly upregulated genes (log2 fold change >1) in each cluster enriched in the gene ontology term of inflammatory response.** The ranking of gene from top to bottom is based on the mean expression level in each cluster.
(TIF)

**S5 Fig. The expression of pro-inflammatory genes in various cell clusters.** (**A**) The highlighted clusters with bright color (i.e. C1, C6, C8, C10, C13, C16) were newly emerged and significantly increased post infection. (**B**) The normalized expression of host 372 genes related to inflammatory response in the significantly infected clusters of different levels of IAV infection (X-axis) (GO: 0006954). The mean expression of each gene was calculated across all cells in the cluster indicated at the Y-axis with $\log2(x+1)$ transformed. The cells in the clusters susceptible to IAV infection were divided into highly infected cells (I), potential or lowly infected cells (P), and uninfected cells (N).
(TIF)

**S6 Fig. The distribution of the number of viral UMI counts per cell in each cluster.** The dots indicate the cells of 18 clusters from different libraries across six time points p.i. with corresponding colors.
(TIF)

**S7 Fig. IAV replication vanished at day 7 post infection.** (**A**) tSNE projection where each cell is colored by log2 sum expression of the eight genes of IAV. (**B**) The mean expression of IAV eight genes in the single cell libraries from lung at different days post infection.
(TIF)

**S8 Fig. Single cell heterogeneity of the intracellular viral load within the susceptible cell clusters of IAV infection.** The cells in the clusters susceptible to IAV infection were divided into highly infected cells (I, total UMI counts of viral transcripts $\geq 8$), potential or lowly infected cells (P, total UMI counts of viral transcripts $\geq 1$), and undetected cells (N, UMI counts of viral transcripts = 0). The percentages of highly infected cells (gray), potential or lowly infected cells (light gray), and undetected cells (dark gray) were shown in y axis. The cell counts of different sub-clusters were shown at the bottom.
(TIF)

**S9 Fig. Heatmap showing the log2 fold change of the pro-inflammatory genes in various cell clusters with high ratios at day 1 p.i..** The hierarchical cluster trees were constructed based on the distance metric of pearson correlation among genes or clusters. The asterisk

indicates the highly expressed pro-inflammatory genes only at day 1 p.i..
(TIF)

**S10 Fig. The graphical representation of some pro-inflammatory genes which were highly-expressed in C13 with tSNE plot of the single cell library at day 1 p.i..** The number at the lower right corner of each box indicates the color up limit for measuring gene expression of single cell.
(TIF)

**S11 Fig. The normalized expression (UMI counts, Y-axis) of the significant genes with high expression related to inflammatory response in the cells of different clusters from different days p.i. (X-axis)**. The dots indicate the cells from different clusters with corresponding colors.
(TIF)

**S12 Fig. The normalized expression (UMI counts, Y-axis) of the significant genes with high expression at day 1 p.i. related to inflammatory response in the cells from C13 (different days p.i., X-axis)**.
(TIF)

**S13 Fig.** Heatmap showing pearson correlation for relationships between the normalized mean expression profiles of cells from different clusters of myeloid-lineage based on the genes related to GO:0006954 inflammatory response (left) and GO:0051607 defensive response to virus (right).
(TIF)

**S14 Fig. The single-cell trajectories of C14-C16-C13 in pseudotime constructed using the package Monocle based on the differential expressed genes among the related single cells.**
(TIF)

**S15 Fig. PD-L1$^+$ neutrophils can be infected by IAV.** (**A**) The relative viral mRNA expression in uninfected cells, PD-L1$^+$ neutrophils, PD-L1$^-$ neutrophils, and T cells from the lung of mice infected with 0.5 LD$_{50}$ of influenza A/PR/8/34 (H1N1) viruses was analyzed with qRT-PCR at day 1 p.i.. Data are shown as the means ± SD in one of three independent experiments. *N.D.* means not detected. (**B**) Immunofluorescent images showing the neutrophils (CD11b$^+$Ly6G$^+$ cells) isolated from lung of mice uninfected (top panel) or infected (bottom panel) with influenza 0.5 LD$_{50}$ of A/PR/8/34 (H1N1) viruses at 24 hours p.i. with anti-FLU-HA (red) and anti-PD-L1 (green) antibodies. The nucleus was stained with DAPI (blue). Scale bars, 40 μm.
(TIF)

**S16 Fig. Heatmap showing the fold change of the pro-inflammatory genes in various cell clusters with high ratios at day 7 p.i..** The hierarchical cluster trees were constructed based on the distance metric of pearson correlation among genes or clusters. The asterisk indicates the highly expressed pro-inflammatory genes only at day 7 p.i..
(TIF)

**S17 Fig. The normalized expression (UMI counts, Y-axis) of the significant pro-inflammatory genes with high expression at day 7 p.i. in the cells of different clusters from different days p.i. (X-axis).** The dots indicate the cells from different clusters with corresponding colors.
(TIF)

**S18 Fig. The normalized expression (UMI counts, Y-axis) of the significant pro-inflammatory genes with high expression at day 7 p.i. in the cells of different clusters from different days p.i. (X-axis).** These selected genes were mainly significantly high express in C6 cluster at day 7 p.i.. The dots indicate the cells from different clusters with corresponding colors.
(TIF)

**S19 Fig. The graphical representation of some pro-inflammatory genes which were highly expressed in C8 cells with tSNE plot of the single cell library at day 7 p.i..** The number at the lower right corner of each box indicates the color up limit for measuring gene expression of single cell.
(TIF)

**S20 Fig.** The normalized expression (UMI counts, Y-axis) of the significant pro-inflammatory genes with high expression at day 7 p.i. in the cells from C8 (different days p.i., X-axis).
(TIF)

**S21 Fig. The expression of Pf4$^+$ cells.** The tdtomato-Pf4 mice were infected with 0.5 LD$_{50}$ of influenza A/PR/8/34 (H1N1) viruses. At day 7 p.i, cells in the lung were stained with anti-Ccl8 antibodies (green), and the nucleus was stained with DAPI (blue). TdTomato-Pf4 was shown in red. Scale bars, left, 100 μm; right, 20 μm. About 60% of Ccl8$^+$ Pf4-tdTomato$^+$ cells were found within Pf4-tdTomato$^+$ cells. At least 300 cells were scored.
(TIF)

**S22 Fig. Several types of cells were analyzed after Pf4$^+$ macrophages depletion.** (**A**) The cell number of neutrophils (CD11b$^+$Ly6G$^+$), interstitial macrophages (CD11b$^+$CD11c$^+$CD64$^{high}$ MHCII$^+$CD24$^-$), and mature monocytes and macrophages (CD11b$^+$MHCII$^{int}$CD24$^-$Ly6C$^+$/ Ly6C$^-$) was analyzed in *Pf4-cre-; iDTR* mice and *Pf4-cre+; iDTR* mice. (**B**) Flow cytometry of Pf4$^+$ cells from the lung sample of Pf4-tdTomato mice infected with IAV. The cells were gated on Pf4-tdTomato$^+$CD11b$^+$. Number in quadrants indicates percent Pf4-tdTomato$^+$CD11b$^+$Ly6C$^+$ cells. (**C**) Left panel: Flow cytometry of infected CD8$^+$ T cells (Flu-NP$^+$CD8$^+$) of lung samples from *Pf4-cre-; iDTR* mice and *Pf4-cre+; iDTR* mice. Numbers in quadrants indicate percent infected CD8$^+$ T cells. Right panel: Frequency of Flu-NP$^+$CD8$^+$ of the lung samples in Left. Data are shown as the means ± SD in one of three independent experiments. ***, $P < 0.001$ (Student t test, n = 5). n.s. means not significant.
(TIF)

**S23 Fig. Circos plots showing the ligand interactions of the immune cells in lung at day 1 p.i. and day 7 p.i..** Line connections indicate the literature supported ligand and receptor interactions. The average UMI counts of all LR genes were log2 transformed, and highlighted with the gradual red color in the graph according the transcription level of LR genes. The LR interaction lines were colored in accordance with the transcription level of ligand or receptor genes in the main cluster.
(TIF)

**S24 Fig. Circos plots showing the receptor interactions of the immune cells in lung at day 1 p.i. and day 7 p.i..** Line connections indicate the literature supported ligand and receptor interactions. The average UMI counts of all LR genes were log2 transformed, and highlighted with the gradual red color in the graph according the transcription level of LR genes. The LR interaction lines were colored in accordance with the transcription level of ligand or receptor genes in the main cluster.
(TIF)

## Acknowledgments

We are grateful to thank Guangzhou Qianyang Biotechnology Co., Ltd and Genedenovo Biotechnology Co., Ltd for assisting in sequencing and bioinformatics analysis.

## Author Contributions

**Conceptualization:** Junsong Zhang, Kai Deng, Hui Zhang.

**Data curation:** Junsong Zhang, Jun Liu, Hui Zhang.

**Formal analysis:** Junsong Zhang, Jun Liu, Jin Wang, Meng Zhao, Gen Lu, Kai Deng, Hui Zhang.

**Funding acquisition:** Junsong Zhang, Feng Huang, Hui Zhang.

**Investigation:** Junsong Zhang, Kai Deng, Hui Zhang.

**Methodology:** Junsong Zhang, Jun Liu, Yaochang Yuan, Baohong Luo, Zhihui Xi.

**Project administration:** Junsong Zhang, Jun Liu, Yaochang Yuan, Feng Huang, Baohong Luo, Ting Pan, Bingfeng Liu, Yiwen Zhang, Xu Zhang, Yuewen Luo, Hui Zhang.

**Resources:** Junsong Zhang, Hui Zhang.

**Software:** Junsong Zhang, Jun Liu, Rong Ma, Zhihui Xi.

**Supervision:** Kai Deng, Hui Zhang.

**Validation:** Junsong Zhang, Jun Liu, Yaochang Yuan.

**Visualization:** Junsong Zhang, Jun Liu, Yaochang Yuan, Feng Huang, Baohong Luo.

**Writing – original draft:** Junsong Zhang, Jun Liu, Feng Huang, Hui Zhang.

**Writing – review & editing:** Junsong Zhang, Jun Liu, Hui Zhang.

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
