## [Decision Letter · Decision Letter 0]

20 Aug 2019

Dear Dr. Zhang,

Thank you very much for submitting your manuscript "Two Waves of Pro-inflammatory Factors Are Released during the Influenza A Virus (IAV)-driven Pulmonary Immunopathogenesis" (PPATHOGENS-D-19-01039) for review by PLOS Pathogens. Your manuscript was fully evaluated at the editorial level and by independent peer reviewers. The reviewers appreciated the attention to an important problem, but raised some substantial concerns about the manuscript as it currently stands. These issues must be addressed before we would be willing to consider a revised version of your study. We cannot, of course, promise publication at that time.

We therefore ask you to modify the manuscript according to the review recommendations before we can consider your manuscript for acceptance. Your revisions should address the specific points made by each reviewer.

(1) A letter containing a detailed list of your responses to the review comments and a description of the changes you have made in the manuscript. Please note while forming your response, if your article is accepted, you may have the opportunity to make the peer review history publicly available. The record will include editor decision letters (with reviews) and your responses to reviewer comments. If eligible, we will contact you to opt in or out.

(2) Two versions of the manuscript: one with either highlights or tracked changes denoting where the text has been changed; the other a clean version (uploaded as the manuscript file).

Additionally, to enhance the reproducibility of your results, PLOS recommends that you deposit your laboratory protocols in protocols.io, where a protocol can be assigned its own identifier (DOI) such that it can be cited independently in the future. For instructions see http://journals.plos.org/plospathogens/s/submission-guidelines#loc-materials-and-methods

We hope to receive your revised manuscript within 60 days. If you anticipate any delay in its return, we ask that you let us know the expected resubmission date by replying to this email. Revised manuscripts received beyond 60 days may require evaluation and peer review similar to that applied to newly submitted manuscripts.

[LINK]

Sincerely,

Andrea J. Sant, Ph.D.

Associate Editor

PLOS Pathogens

Ron Fouchier

Section Editor

PLOS Pathogens

Kasturi Haldar

Editor-in-Chief

PLOS Pathogens

orcid.org/0000-0001-5065-158X

Grant McFadden

Editor-in-Chief

PLOS Pathogens

orcid.org/0000-0002-2556-3526

This is an intriguing and novel set of studies that point to temporally distinct waves of proinflammatory mediators and suggests a network of interactions in the inflammatory components during the course of an immune response to influenza. There was clearly considerable effort in this study, which is to be commended. Despite these strengths, there are substantial areas of concern expressed by each reviewer, particularly reviewers 2 and 3, that require attention, at multiple levels, including data analyses, clarity of the explanation and rationale of the experiments performed and finally, additional experimentation. The authors should read the thoughtful reviews carefully and address these points and only after these issues are addressed should the authors consider submitting a revision.

Reviewer's Responses to Questions

**Part I - Summary**

Reviewer #1: The authors present a very interesting study examining the inflammatory response in different cell types over time.

One major weakness is the n=3 mice per groups and number of cells examined (sometimes less than 100)

Reviewer #2: The manuscript by Zhang and colleagues details a single cell RNASeq study of mouse lung before and following influenza infection over 5 time points. In general, the presentation of the vast data set is beautifully executed with cutting edge bioinformatic approaches. The authors find numerous interesting data including that viral load in cells did not impact inflammatory cytokine production, that early inflammatory gene expression is mostly from Pdl1+ neutrophils, and that later inflammatory gene expression tracks with pf4+ macrophages. The authors follow up on the pf4+ macrophage finding using a DTR system to delete these cells and assess inflammation endpoints. Interactome data at the end of the study reveal several autocrine loops that drive inflammation. While the concept that neutrophils drive early and macrophages late inflammation in response to influenza infection is not new, the resolution of the kinetic data presented is novel and very interesting to the field. The study could be improved by follow up regarding the Pdl1+ PMN and addition of data regarding interferon producing cells.

Reviewer #3: Understanding of the dynamics and coordination of the immune response to influenza A virus infection is important for biomarker discovery and identification of targets for intervention. This study uses the murine influenza model and single-cell RNA sequencing to analyse the changes in pulmonary cellular composition over the first 12 days of infection. At day 0, 1, 3, 5, 7 and 12 post-infection, lung cell suspensions from 4 mice were mixed and analysed by 10x Chromium. At each time-point ~2000-3300 individual cells underwent scRNA-seq. By tSNE, cells were clustered and clusters categorised into cell type by their expression of signature genes. A Gene Ontogeny module was used to probe each cluster at every time-point to identify those clusters with high expression of inflammatory response genes. One cluster was associated with early inflammatory response at day 1 and this was found to be PD-L1 expressing neutrophils; while another cluster associated with inflammatory response at day 7 was found to be PF4 (or CXCL4) expressing macrophages. Additionally, some correlation analysis was performed to identify associations between chemokine/cytokine receptors and their ligands expressed in different clusters.

Overall, the manuscript is confusingly written, with the rationale for each analysis incompletely expressed and many essential details overlooked. In many cases, supplementary data are cited but not explained in the Results. Some experiments are not adequately explained so interpretation is often difficult. The study is basically descriptive but the authors claim to identify mechanisms of pathogenesis. There are generally issues with the English and the paper would benefit from a thorough review of the language.

**Part II – Major Issues: Key Experiments Required for Acceptance**

Reviewer #1: The authors have examined groups of 3 mice at multiple timepoints. This is relatively low and may not represent variability. In addition, different types of cells were analysed however for some groups less than 100 cells were examined. This cell numbers are very low. It was not clear what the authors have done to account for this?

A t test was used throughout the manuscript. This is not appropriate when comparing multiple groups.

Some of the figures are very small and axis labels are hard to read eg Fig 3B

Reviewer #2: 1) Mechanistic Studies – The authors conducted mechanistic studies to confirm the role of pf4+ macrophages. Did depletion of these cells impact PMN of other macrophage populations? How specific was the depletion method utilized for pf4+ cells alone (beyond the data showing no impact on T cells). The authors should target pdl1+ PMN early during infection to generate similar data implicating these cells as critical inflammatory drivers.

2) IFN Producing Cells – As it is known that type I, II, and III IFNs regulate inflammation during influenza infection, it would be helpful to highlight data showing which cells make these antiviral cytokines during the infection time course.

3) Tissue Dissociation – The authors observed almost no epithelial cells in this study. They should comment on how the single cell suspension methodology appears to have selected for leukocytes and how this limits interpretation in the discussion.

Reviewer #3: Line 163. The GO term 0006954 was chosen to identify clusters making an inflammatory response. This is a very biased approach. How was this GO term chosen over the many others? The authors should justify this strategy and discuss its limitations. Could a more unbiased strategy be used?

Line 169. In this analysis, the epithelial cell cluster C18 has a much lower proportion of IAV transcript-expressing cells than monocyte/macrophage and granulocyte clusters. Can this be correct? Productive IAV replication in vivo is unlikely in monocytes and neutrophils, while epithelial cells are the main target for IAV. How can this be explained? Could there be a technical reason for this analysis to be skewed? More explanation of Figures S6, S7 and Table S6 is necessary.

Line 172. This is unexpected. Surely more viral components ought to trigger greater PRR signalling? Again, could there a technical reason for the expression counts of IAV transcripts not to be correct?

Line 310. “high level of virus infection in C13-neutrophils was revealed” This should be validated. Could viral transcripts be quantified from sorted cells by qPCR?

Line 233. Does depleting Pf4-expressing cells only affect this population of macrophages? What effect does it have on CXCL4-induced chemotaxis, if any? The cited reference is regarding megakaryotes. More data on the model in the context of the lung is required to support the authors' interpretation.

**Part III – Minor Issues: Editorial and Data Presentation Modifications**

Reviewer #1: Can the dose of virus given to the mice be represented as PFU to allow comparison to other studies.

Reviewer #2: 1) Supp Figures – The description of the data in the numerous Supp Figures is sparse or not present. Particularly for Supp Fig8-12 and 15-18, these are glossed over in the results section.

2) Grammar – There are some minor grammatical issues in the text.

Reviewer #3: Line 144. Clusters were categorised according to “known markers of major cell types”. How were these chosen and validated?

Line 152. “The complex intrinsic components of cells…” What does this mean?

Line 165. The phrase “exerted strong pro-inflammatory response” is not an accurate reflection of what the method shows; this is a transcriptional signature, not a functional readout. Suggest re-phrasing.

Line 173. “indicated the roles of innate immune cells in controlling IAV-driven pneumonia” The analysis has not shown the roles only described the presence and changing composition of immune cells. Sugges re-phrase.

Line 186. Supplementary figures have not been explained.

Line 187-9. The correlation analysis shows correlation between several clusters that have been labelled by the authors as granulocyte-like but does not actually show granulocyte characteristics as claimed.

Line 191. Where did the data from eosinophils and basophils come from?

Line 192. “further confirmed by” The pseudotime ordering analysis suggests but does not confirm the transition between clusters. Suggest re-phrase.

Line 201 & Figure 2H. What does “much higher” mean? Where are the statistical tests shown?

Line 203. What does “super-activated” and “cytokine-hypersecreted” mean and in relation to what? Also, only cytokine transcripts are shown, not protein. Suggest re-phrase.

Line 204. Only viral transcripts are shown, no “virus protein HA”. Suggest re-phrase.

Line 227. Error in citing figures: 3F is a correlation analysis, 3G is the PCA analysis and there is no GO enrichment analysis shown.

Line 241. It is unclear how the experiments in Figure 4A and 4B were performed. The figure legend says that RT-PCR was done in Pf4+-macrophages but these are ablated in the Pf4-cre+ mice. There is nothing about ELISA in the materials and methods, and it is unclear what sample type CCL8 and CCL2 have been measured in.

Line 255. “indicated that Pf4-positive macrophages were the precursors of AMs” I do not think the data show this. Reduction in AMs could be indirect. Suggest re-phrase

Line 285. “further confirmed the interaction” I do not think the data actually confirms these interactions since it only shows correlations.

Line 304. “play a leading role” The authors have shown these cells expressing inflammation-related genes at day 1 but not actually demonstrated their role.

Line 370. “demonstrated many new mechanisms” I do not think any new mechanisms have been demonstrated. Sequential waves of innate and adaptive immune cells have been shown in many respiratory viral infection systems. Although neutrophils are thought of as anti-bacterial, they have also been implicated in antiviral responses. I think the authors are over-claiming.

Materials and methods. The method for ELISA should be described.

Figure 2A: The symbols # and ## are too small to see.

Figure 2A legend: “# indicated the highly-expressed genes” Do # and ## indicate statistically significant differentially expressed genes? Suggest re-phrasing legend to clarify.

Figure 2B legend: “normalized expression” Normalised how and to what?

Figure 2C legend: “were unregulated” What does this mean?

Figure S5 legend. “The highlighted clusters…” There are no highlighted clusters. Later in the legend, there is repetition of a sentence.

Figure S7. This and plots like it (e.g. S17) are almost uninterpretable. There are too many colours that are poorly explained and the superimposed dots make it impossible to appreciate the time-line.

Figure S10. How have these pro-inflammatory genes here (other representative genes in other figures e.g. S11, S12, S16) been selected?

Figure S18. Please provide statistical test results to show significant changes over time.

Figure S19. This is not very convincing. Is there any quantitative co-expression analysis available?

Figure S20. Please provide a plot summarising frequencies between groups and statistics.

PLOS authors have the option to publish the peer review history of their article (what does this mean?). If published, this will include your full peer review and any attached files.

Reviewer #1: No

Reviewer #2: Yes: John F Alcorn

Reviewer #3: No

---

## [Decision Letter · Decision Letter 1]

14 Nov 2019

Dear Dr. Zhang,

Thank you very much for submitting your manuscript "Two Waves of Pro-inflammatory Factors Are Released during the Influenza A Virus (IAV)-driven Pulmonary Immunopathogenesis" (PPATHOGENS-D-19-01039R1) for review by PLOS Pathogens. Your manuscript was fully evaluated at the editorial level and by independent peer reviewers. The reviewers appreciated the attention to an important problem, but raised some substantial concerns about the manuscript as it currently stands. These issues must be addressed before we would be willing to consider a revised version of your study. We cannot, of course, promise publication at that time.

We therefore ask you to modify the manuscript according to the review recommendations before we can consider your manuscript for acceptance. Your revisions should address the specific points made by each reviewer.

(1) A letter containing a detailed list of your responses to the review comments and a description of the changes you have made in the manuscript. Please note while forming your response, if your article is accepted, you may have the opportunity to make the peer review history publicly available. The record will include editor decision letters (with reviews) and your responses to reviewer comments. If eligible, we will contact you to opt in or out.

(2) Two versions of the manuscript: one with either highlights or tracked changes denoting where the text has been changed; the other a clean version (uploaded as the manuscript file).

Additionally, to enhance the reproducibility of your results, PLOS recommends that you deposit your laboratory protocols in protocols.io, where a protocol can be assigned its own identifier (DOI) such that it can be cited independently in the future. For instructions see http://journals.plos.org/plospathogens/s/submission-guidelines#loc-materials-and-methods

We hope to receive your revised manuscript within 60 days. If you anticipate any delay in its return, we ask that you let us know the expected resubmission date by replying to this email. Revised manuscripts received beyond 60 days may require evaluation and peer review similar to that applied to newly submitted manuscripts.

[LINK]

Sincerely,

Andrea J. Sant, Ph.D.

Associate Editor

PLOS Pathogens

Ron Fouchier

Section Editor

PLOS Pathogens

Kasturi Haldar

Editor-in-Chief

PLOS Pathogens

orcid.org/0000-0001-5065-158X

Grant McFadden

Editor-in-Chief

PLOS Pathogens

orcid.org/0000-0002-2556-3526

The authors have made a serious effort to respond to the criticisms of the first submission. This second submission is much improved. Reviewer 3 raises important issues of clarity and suggestions for more complete analyses of data and serious attention must be paid to these issues and suggestions.

Reviewer's Responses to Questions

**Part I - Summary**

Reviewer #1: The authors have not addressed my concern of n=3 for the mouse studies. Could major findings be validated with additional experiments?

Reviewer #2: The authors have carefully addressed my concerns and I think the utility of the data for the field is high.

Reviewer #3: Overall, the manuscript has improved substantially. It certainly presents a large body of work and has sought to validate the transcriptomics findings with a variety of follow-up studies. The findings are interesting and the dataset represents a useful resource. The flow and readability of the paper are much better, although there still remain some grammatical errors here and there.

**Part II – Major Issues: Key Experiments Required for Acceptance**

Reviewer #1: (No Response)

Reviewer #2: (No Response)

Reviewer #3: Line 249: the authors mention and dismiss the C6 cluster as a source of pro-inflammatory cytokine expression. Looking at their data, these are lymphocytes and likely to represent CD8+ T cells, which would be expected to be expanded and recruited at the day 7 timepoint. Ideally, this population would be analysed in more detail as their contribution is not negligible. The authors may feel that this is beyond the scope of the paper, given that they have focused solely on innate cells, but additional analysis of this population would provide a more complete view and increase the impact. The importance of other subsets is further supported by the lack of complete abrogation following depletion of the Pf4+ macrophages (Fig 4C-D).

**Part III – Minor Issues: Editorial and Data Presentation Modifications**

Reviewer #1: (No Response)

Reviewer #2: (No Response)

Reviewer #3: Line 56: this implies that pandemics are yearly. Suggest rephrase.

Line 61: "we characteristically monitored..." The meaning of this is unclear. Suggest rephrase.

Line 63: "resources for the release..." The meaning of this is unclear. Suggest rephrase.

Lines 121 "high-quality" and 127 "the most stable and accurate". Suggest removing this qualifiers as they are disputable.

Line 143 "Some significant genes..." This remains unclear. Do the authors mean that all genes meeting their statistical criteria are shown or that only a selected subset, and if the latter, what were the selection criteria?

Line 187-189. From the response to reviewers, the authors suggest that this lack of correlation is due to inflammatory activation by cytokines and other bystander effects. It would be worth discussing this in more detail in the Discussion.

Figure 4G is still extremely difficult to interpret. Suggest a plot showing data from the C8 cluster alone.

PLOS authors have the option to publish the peer review history of their article (what does this mean?). If published, this will include your full peer review and any attached files.

Reviewer #1: No

Reviewer #2: No

Reviewer #3: No

---

## [Editor Report · Decision Letter 2]

8 Jan 2020

Dear Dr. Zhang:

Thank you very much for submitting your manuscript "Two Waves of Pro-inflammatory Factors Are Released during the Influenza A Virus (IAV)-driven Pulmonary Immunopathogenesis" (PPATHOGENS-D-19-01039R2) for review by PLOS Pathogens. Your manuscript was fully evaluated at the editorial level and by independent peer reviewers. The reviewers appreciated the attention to an important topic but identified some aspects of the manuscript that should be improved.

We therefore ask you to modify the manuscript according to the review recommendations before we can consider your manuscript for acceptance. Your revisions should address the specific points made by each reviewer.

(1) A letter containing a detailed list of your responses to the review comments and a description of the changes you have made in the manuscript. Please note while forming your response, if your article is accepted, you may have the opportunity to make the peer review history publicly available. The record will include editor decision letters (with reviews) and your responses to reviewer comments. If eligible, we will contact you to opt in or out.

(2) Two versions of the manuscript: one with either highlights or tracked changes denoting where the text has been changed; the other a clean version (uploaded as the manuscript file).

We hope to receive your revised manuscript within 60 days or less. If you anticipate any delay in its return, we ask that you let us know the expected resubmission date by replying to this email.

[LINK]

Sincerely,

Andrea J. Sant, Ph.D.

Associate Editor

PLOS Pathogens

Ron Fouchier

Section Editor

PLOS Pathogens

Kasturi Haldar

Editor-in-Chief

PLOS Pathogens

orcid.org/0000-0001-5065-158X

Michael Malim

Editor-in-Chief

PLOS Pathogens

orcid.org/0000-0002-7699-2064

The manuscript is much improved and the added data through these revisions have greatly improved the value of these experiments to the field. This is a very interesting and provocative study.

In the response to Reviewer #1's issue regarding the number of samples analyzed and respond in the response to review but did not add the data shown in "Figure 1" in response to the review to the actual manuscript that generally addresses the issue raised by the reviewer. They should include it as supplementary data with more information on how these experiments were performed.

---

## [Editor Report · Decision Letter 3]

19 Jan 2020

Dear Dr. Zhang,

We are pleased to inform you that your manuscript 'Two Waves of Pro-inflammatory Factors Are Released during the Influenza A Virus (IAV)-driven Pulmonary Immunopathogenesis' has been provisionally accepted for publication in PLOS Pathogens.

Before your manuscript can be formally accepted you will need to complete some formatting changes, which you will receive in a follow up email. A member of our team will be in touch within two working days with a set of requests.

Best regards,

Andrea J. Sant, Ph.D.

Associate Editor

PLOS Pathogens

Ron Fouchier

Section Editor

PLOS Pathogens

Kasturi Haldar

Editor-in-Chief

PLOS Pathogens

orcid.org/0000-0001-5065-158X

Michael Malim

Editor-in-Chief

PLOS Pathogens

orcid.org/0000-0002-7699-2064

This interesting manuscript describes intriguing data that supports that conclusions made by the authors. It has been much improved through the revision process.
---

## [Editor Report · Acceptance letter]

20 Feb 2020

Dear Dr. Zhang,

We are delighted to inform you that your manuscript, "Two Waves of Pro-inflammatory Factors Are Released during the Influenza A Virus (IAV)-driven Pulmonary Immunopathogenesis," has been formally accepted for publication in PLOS Pathogens.

Best regards,

Kasturi Haldar

Editor-in-Chief

PLOS Pathogens

orcid.org/0000-0001-5065-158X

Michael Malim

Editor-in-Chief

PLOS Pathogens

orcid.org/0000-0002-7699-2064